# Calcium scoring during 18F-FDG PET/CT in cancer indications: Improving cardiovascular risk stratification and prevention

Réjane Mazet[1], Nouritza Torossian[2], Mathilde Wanneveich[3], Gilles Metrard[1], Angela Malmare[4], Denis Angoulvant[5,6], Matthieu Bailly[1,7]*

1 Nuclear Medicine Department, CHU ORLEANS, Orleans, France, 2 Oncology Department, CHU ORLEANS, Orleans, France, 3 Research Department, CHU ORLEANS, Orleans, France, 4 Cardiology Department, CHU ORLEANS, Orleans, France, 5 Cardiology Department, CHRU TOURS, Tours, France, 6 ISCHEMIA, INSERM U1327, Tours University, Tours, France, 7 LI2RSO, Orleans University, Orleans, France

* matthieu.bailly@chu-orleans.fr

## Abstract

### Purpose

Coronary artery calcium (CAC) can be assessed on 18F-FDG PET/CT in oncology patients. We evaluated the proportion of oncology patients referred for PET/CT imaging in whom systematic CAC scoring could reclassify cardiovascular (CV) risk and modify primary cardiovascular disease (CVD) prevention strategies.

### Materials and methods

This prospective study included 190 cancer patients undergoing 18F-FDG PET/CT in oncological indications at our institution between June and July 2024. Data on CV risk factors, personal/family CVD history, and current treatments were collected. CAC scores were assessed visually and automatically on CT scans. Patients' CV risk was reclassified according to CAC scores and compared with their current lipid-lowering therapy (LLT) and cardiology follow-up.

### Results

A total of 190 subjects were analyzed (mean age 65 ± 13 years, 64% female, mean Framingham Risk Score 17.2 ± 9.2%). Patients were categorized as follows according to CAC: 24.7% (N = 47) no CAC, 36.8% (N = 70) mild, 21.6% (N = 41) moderate, and 16.8% (N = 32) severe CAC. Among them, 56.3% (N = 107) had a previous consultation with a cardiologist, 30.0% (N = 57) were on LLT, and 6.8% (N = 13) had a history of CVD. Based on CAC scoring, 43% (N = 81) could be reclassified in a different CVD risk category and 49% (N = 93) required changes in primary CVD prevention through LLT adjustments or initiating follow-up for high risk of CVD.

**Data availability statement:** The data underlying the results presented in the study are available from Direction Recherche Innovation of CHU Orleans (recherche.clinique@chu-orleans.fr).

**Funding:** The author(s) received no specific funding for this work.

**Competing interests:** The authors have declared that no competing interests exist.

**Abbreviations:** ACC, American College of Cardiology; AHA, American Heart Association; CAC, Coronary Artery Calcium; CAD, Coronary Artery Disease; CHD, Coronary Heart Disease; CT, Computed Tomography; CV, Cardiovascular; CVD, Cardiovascular Disease; ECG, Electrocardiogram; ESC, European Society of Cardiology; FRS, Framingham Risk Score; HU, Hounsfield Unit; LLT, Lipid-Lowering Therapy; PET, Positron Emission Tomography; SD: Standard Deviation.

## Conclusion

Systematic CAC assessment on PET/CT imaging could enhance CV risk stratification and prevention in oncology patients.

## Introduction

PET/CT is now considered standard of care in oncology. Many cancer patients are at increased risk of cardiovascular disease (CVD) due to shared risk factors, such as age, smoking and cardiotoxicity of cancer treatments [1,2]. Consequently, cancer patients have an increased risk of cardiovascular (CV) mortality compared to the general population [3].

CVD has been the leading cause of death in the United States every year since 1919, ahead of cancer, with coronary artery disease (CAD) being the most common cause both in the general population and in elderly women [4]. It is important to assess CV risk in cancer patients as it may impact both their treatments and prognosis. Tools such as the SCORE and Framingham Risk Score (FRS) help estimate individual patient's risk of future CV events. Coronary Artery Calcium (CAC) scoring through computed tomography (CT) provides additional risk stratification and is particularly useful in reclassifying patients categorized as "intermediate risk" by clinical and biological scoring systems [5,6]. The Agatston score, measured using ECG-gated, non-contrast cardiac CT scans, is the gold standard for CAC assessment but involves additional radiation exposure. Studies showed that low-dose, non-ECG-gated CT scans can also accurately stratify patients according to CV risk [7] and artificial intelligence may help analyze these scans on a large-scale [8,9].

Thus, it may be possible to systematically assess CAC in asymptomatic patients, potentially at-risk of CV events, referred for PET/CT cancer imaging [10]. However, the correlation between CV risk estimated by FDG PET-derived CAC and clinical prevention strategies (such as cardiology follow-up or statin prescription) remains poorly studied in real-world cancer populations.

In this study, we prospectively assessed CAC of patients referred for 18F-FDG PET/CT cancer imaging and compared it with their CV history, medications, and cardiac follow-up.

## Materials and methods

### Patient population

From June 2024 to July 2024, 200 patients referred for 18F-FDG PET/CT were prospectively enrolled in the CalcoTEP trial (clinicaltrials.gov unique identifier NCT06379295). Inclusion criteria were 18F-FDG PET/CT imaging. Exclusion criteria included patients referred for brain 18F-FDG PET/CT imaging, patients already enrolled in this study, pregnant or breastfeeding women, minors, and protected adults. Of these patients, we included in this study 190 patients referred for cancer investigation.

CV risk factors, history of CVD and current medications were collected using a standardized self-report questionnaire.

Every patient received information regarding the study and gave verbal informed consent in the presence of an investigator, which was documented in the medical record. The study protocol was approved by the Local and Regional Ethics Committees (CPP SUD-OUEST ET OUTRE MER III) and the procedures were in accordance with the Declaration of Helsinki. No technical issues were encountered; all CT scans acquired alongside the PET imaging were of sufficient quality to allow for CAC assessment.

### PET/CT imaging

As standard of care, patients underwent PET/CT imaging one hour after 18F-FDG injection (including a free-breathing, ungated low-dose CT scan). Images were acquired in 3D mode on a Biograph mCT™ 64-slice PET system (Siemens Healthineers, Germany) or a Vereos™ 64-slice PET system (Philips, Cleveland OH). CT scan could be acquired with contrast-enhancement, depending on the clinical indication.

### Coronary calcium score evaluation

Using the low-dose non-electrocardiogram-gated CT scan performed in conjunction with the PET acquisition, the CAC score was estimated as the sum of the CAC score content in the three main coronary arteries using the method described by Agatston et al [11]. This assessment was performed visually [12] and using syngo.CT CaScoring™ (Siemens Healthineers, Germany). CAC quantification was threshold based. For non-contrast CT scans, the original lower threshold of 130 HU was used. For contrast-enhanced CT scans, a region of interest (ROI) was placed in the ascending aorta. The mean attenuation and the standard deviation (SD) in Hounsfield Units (HU) were collected. The lower threshold for determining the presence of CAC was defined as follows: threshold = mean aortic (HU) + 2 SD (HU) (Fig 1) [13].

In this study, the assessment was independently performed by two physicians (a junior and a senior expert), with a qualitative assessment in four groups that correlates with traditional CAC score groups: 0 (no CAC), 1–99 (mild), 100–399 (moderate) and CAC ≥ 400 (severe) [14]. Patients were then reclassified as low (no CAC), intermediate (mild CAC) or high-risk (moderate and severe CAC) of CVD based on their CAC, according to 2019 American College of Cardiology (ACC) and American Heart Association (AHA) guidelines [15].

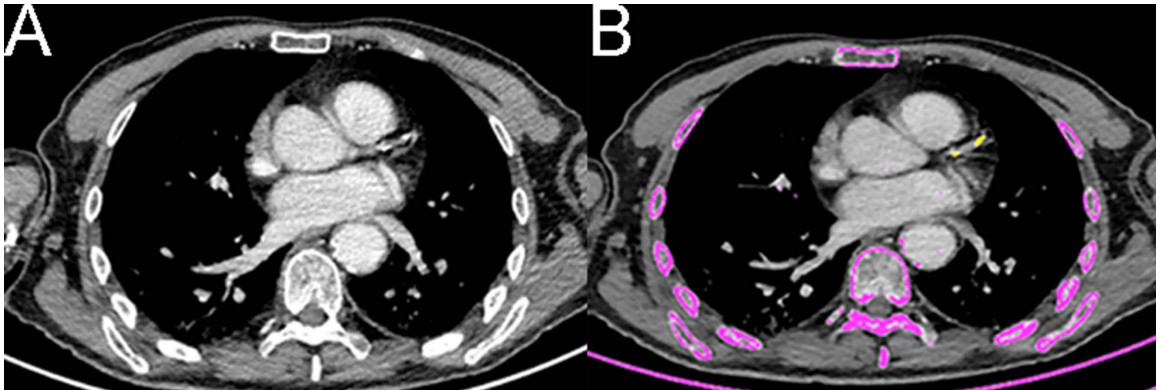

**Fig 1. Coronary artery calcium score assessment. A** Low-dose contrast-enhanced CT scan performed in association with PET acquisition. **B** Automatic contouring (in pink) of pixels with density above a custom threshold defined according to the density in the ascending aorta. Manual selection of coronary calcification lesions, highlighted in yellow for the left anterior descending (LAD) artery on this slice.

## Evaluation of risk factors for cardiovascular disease

Patients were identified as having diabetes, hypertension, or dyslipidemia based on self-reported use of anti-diabetic, antihypertensive, or lipid-lowering therapy (LLT), respectively.

A history of coronary heart disease (CHD) was defined as a self-reported history of myocardial infarction or coronary artery revascularization either by stent implantation or bypass surgery.

Patients self-reported their smoking status as current, former, or never smokers, along with the duration of smoking. Current and former smokers also provided their average daily cigarette consumption. Pack-years were then calculated using the reported daily cigarette intake and smoking duration. Based on patients self-reported data, we calculated an adapted FRS for the 10-year risk of fatal and non-fatal CVD, according to previously published formulae [16,17]. The 10-year CVD risk was categorized in 3 groups: low risk (< 10%), intermediate risk (10–19%) and high risk (>20%).

## Clinical implications of CAC

The management of patients for primary prevention of CVD based on their CAC score was compared with the 2019 ACC guidelines: for patients with a CAC score of 0, risk should be downgraded. In case of no additional high-risk factors such as diabetes mellitus, family history of CHD, or smoking, the guidelines also recommend withholding statin therapy. For CAC subgroups 1–99, statin therapy may be considered for those aged ≥ 55 or those with higher-risk conditions. Patients with a CAC score ≥ 100 should be reclassified into a high-risk group for CV events and require regular follow-up as well as statin therapy [15].

Moreover, while no specific therapeutic adjustments are recommended for younger patients (≤ 45 years), studies have shown that even a low CAC score in this population is a significant marker for increoneased risk of developing obstructive CAD leading to CHD [18]. In these patients, a CAC > 0 automatically places them above the 75th percentile for their age group, requiring LLT.

As such, we sought to identify the proportion of patients within these categories who might benefit from increased attention or potential changes in preventive treatment strategies.

## Statistical analysis

Categorical variables were expressed as numbers and relative frequencies (percentages). Continuous variables were reported as mean ± standard deviation (SD) or median (IQR).

Interobserver agreement was assessed using Cohen kappa coefficient. The differences in median CAC among risk factor subgroups were evaluated using Wilcoxon rank-sum tests and Benjamini-Hochberg method for multiple comparisons adjustments. Student's t-test was performed to study the potential age difference between compared risk factors subgroups. The Spearman correlation test was used to measure the degree of association between CAC score, 10-year FRS and age. The association between the indication for PET/CT imaging and the presence of at least moderate CAC was assessed using Fisher exact test. All tests were two-tailed and $P$ values of < 0.05 were considered to indicate statistical significance. Participants with missing data or those undergoing PET/CT imaging for non-oncologic indications had been excluded from the analyses.

## Results

### CAC results

A total of 190 patients undergoing 18F-FDG PET/CT for cancer investigation (mean age 65 ± 13 years, 64% female) were enrolled (Table 1). 17 (9%) patients were ≤ 45 years. Of the total, 107 (56%) patients had a previous consultation with a cardiologist, 57 (30%) were on at least one LLT and 13 (7%) had a history of CVD. 3 (2%) patients reported a history of invasive coronary angiography while having no subsequent cardiology follow-up. Quantitative CAC scoring was feasible in

**Table 1. Patient demographics.**

| | |
|---|---|
| Sex, female | 121 (63.7) |
| Age (years) | 64.9 ± 13.3 |
| CV risk factors (n = 157) | |
| Diabetes mellitus | 27 (14.2) |
| Hypertension | 80 (42.1) |
| Dyslipidemia | 57 (30.0) |
| Family history of CHD | 59 (31.1) (no data: 1) |
| Smoking | |
| Never | 102 (53.7) |
| Previous | 66 (34.7) |
| Current | 22 (11.6) |
| Pack-years (n = 94) | 16.3 ± 17.6 (no data: 3) |
| FRS (%) | 17.2 ± 9.2 |
| Indication | |
| Hematologic | 35 (18.4) |
| Gynecologic | 74 (38.9) |
| Breast cancer | 58 (30.5) |
| Head and neck | 11 (5.8) |
| Lung | 11 (5.8) |
| Other cancer | 59 (31.1) |
| CAC (Agatston units) | 34 (1-170) |
| CAC category (n = 190) | |
| CAC 0 (no calcium) | 47 (24.7) |
| Mild | 70 (36.8) |
| Moderate | 41 (21.6) |
| Severe | 32 (16.8) |

Data for continuous variables are presented as median (IQR) or mean ± SD. Data for categorical variables are presented as number (percentage).

CAC, coronary artery calcium; CHD, coronary heart disease; CV, cardiovascular; FRS, Framingham risk score.

all patients except one who was unable to maintain a strict supine position. 137 (72%) CT scans were contrast-enhanced. Mean FRS was 17.2 ± 9, with 50 (26%) participants in the low-risk category, 67 (35%) in intermediate-risk and 72 (38%) in the high-risk category. The prevalence of any CAC was 75%. Based on CAC assessment, patients were categorized into four groups: 47 (25%) had no detectable CAC, 70 (37%) had mild CAC, 41 (22%) had moderate CAC, and 32 (17%) had severe CAC. There was no significant difference between the visual assessment of coronary calcifications and the quantitative CAC score evaluation. Interobserver agreement was excellent for visual ordinal CAC scores ($\kappa = 0.90$).

Median CAC score was significantly higher in male patients ($p < 0.001$), in subgroups with personal history of CVD ($p < 0.001$), diabetes mellitus ($p = 0.002$), hypertension ($p < 0.001$) and dyslipidemia ($p < 0.001$). There was no significant difference among patients with personal history of smoking (former and active) ($p = 0.08$) or a family history of CHD ($p = 0.83$) (S1 Table).

CAC score was moderately correlated with FRS ($\rho = 0.60$, $p < 0.001$) and with patients' age ($\rho = 0.53$, $p < 0.001$).

## CAC according to cardiovascular history

Regarding the potential redistribution of subjects into different FRS categories based on CAC scoring, 40 (21%) subjects had a downgrade of their estimated risk and 42 (22%) had an upgrade of their estimated risk (Fig 2).

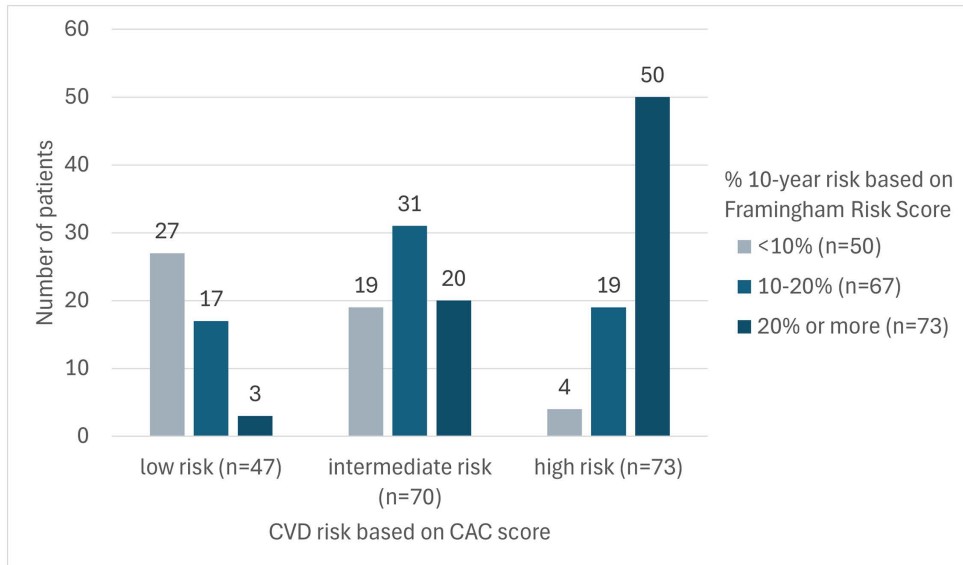

**Fig 2. Distribution of patients with % 10-year risk based on FRS vs CAC scoring.** CAC-based risk categories are displayed on the x-axis, while risk categories based on FRS are shown as column groups. The figure illustrates the reclassification of patients' cardiovascular risk when CAC scoring is applied in comparison with FRS. CAC, coronary artery calcium; CVD, cardiovascular disease; FRS, Framingham risk score.

Among the 80 patients with no previous consultation with a cardiologist and no previous history of CVD or invasive coronary angiography, 21 (26%) had moderate or severe CAC, requiring regular follow-up. Among patients not receiving LLT, 41 (31%) patients had moderate or severe CAC. An additional 42 (32%) patients had mild CAC and were either aged ≥ 55 or had at least one other high-risk condition such as active smoking, diabetes mellitus or a family history of CHD. Among patients younger than 55 with no other high-risk conditions, 7 (5%) had mild CAC.

Among those on LLT, one patient, a non-smoker without diabetes or personal or family history of CHD, had no detectable CAC (2%) making him a candidate for statin therapy discontinuation according to guidelines.

Among patients younger than 45, 6 (35%) had detectable CAC, warranting regular follow-up and possibly initiating statin therapy (Table 2).

Overall, 43% of participants could experience a reclassification of their FRS category – either upward or downward – based on CAC scoring, and 49% would warrant modifications in their CVD primary prevention strategy, including initiating or discontinuing LLT or starting routine follow-up indicated for individuals at high CVD risk (Fig 3).

## CAC according to cancer type

In subgroup analysis by PET/CT imaging indication, 8 of 11 lung cancer patients (73%) had moderate or severe CAC, representing the group with the highest prevalence of elevated CAC scores (p = 0.008) (S1 Fig). 74 patients were referred for gynecologic cancer, including 58 for breast cancer.

Among those with a history of breast cancer (Table 3) (mean age 63 ± 13 years, all female), all were over 45 years old. Mean FRS was 12.4 ± 7, with 25 (43%) in low-risk category, 22 (38%) in intermediate-risk and 11 (19%) in the high-risk category.

Of all breast cancer patients, 35 had a previous consultation with a cardiologist, 15 were on LLT, and 2 had a history of CVD. 13 patients had no CAC, 31 had mild CAC, 10 had moderate CAC, and 4 had severe CAC. Comparing CAC scoring and FRS, 7 (12%) subjects had a downgrade of their estimated risk and 22 (38%) had an upgrade of their estimated risk.

**Table 2. CAC according to clinical findings.**

| Patients who had never consulted a cardiologist and had no history of CVD or invasive coronary angiography | 80 |
|---|---|
| CAC 0 | 25 (31.3) |
| Mild CAC | 34 (42.5) |
| Moderate CAC | 13 (16.3) |
| Severe CAC | 8 (10.0) |
| Patients not on LLT | 133 |
| CAC 0 | 43 (32.3) |
| Mild CAC, aged < 55 and no higher-risk condition | 7 (5.3) |
| Mild CAC and aged ≥ 55 or at least one higher-risk condition | 42 (31.6) |
| Moderate CAC | 23 (17.3) |
| Severe CAC | 18 (13.5) |
| Patients on LLT | 57 |
| CAC 0 and no higher-risk condition | 1 (1.8) |
| CAC 0 and higher-risk condition | 3 (5.3) |
| Mild CAC | 21 (36.8) |
| Moderate CAC | 18 (31.6) |
| Severe CAC | 14 (24.6) |
| Patients ≤ 45 years | 17 |
| CAC 0 | 11 (64.7) |
| At least one CAC | 6 (35.3) |

Data are presented as number (percentage).

CAC, coronary artery calcium; CVD, cardiovascular disease; LLT, lipid-lowering therapy.

Among 22 breast cancer patients who never had a consultation with a cardiologist and had no history of CVD or invasive coronary angiography, 4 (18%) had moderate or severe CAC indicating a regular follow-up.

Among breast cancer patients not on LLT, 8 (19%) had moderate or severe CAC. An additional 17 (40%) patients had mild CAC and were either aged ≥ 55 or had at least one other high-risk condition. Therefore, among breast cancer patients referred for PET/CT imaging who were not on LLT, 58% may be candidates for statin therapy initiation.

In summary, in patients with history of breast cancer, 50% would potentially have an upgrade or downgrade of FRS category based on CAC scoring and 48% would require adjustments regarding primary prevention for CVD, either through the initiation of regular follow-up for high CVD risk and/or changes in LLT.

## Discussion

Our study shows that 26% of patients referred for oncological 18F-FDG PET/CT who never had a consultation with a cardiologist had moderate to severe CAC, warranting regular follow-up. Additionally, 62% of patients not receiving LLT could be candidates for statin therapy according to ACC guidelines. Among patients under 45, 35% had at least one CAC, indicating higher CV risk compared to the general population and suggesting that they may benefit from regular follow-up and statin therapy. In total, 49% of patients required adjustments in primary prevention for CVD (initiation of regular follow-up for high risk of CVD and/or changes in LLT).

Our findings suggest that male patients and those with diabetes, hypertension and dyslipidemia have a higher median CAC score and that there is a significant correlation between age or FRS and CAC score, in accordance with previous studies [19,20].

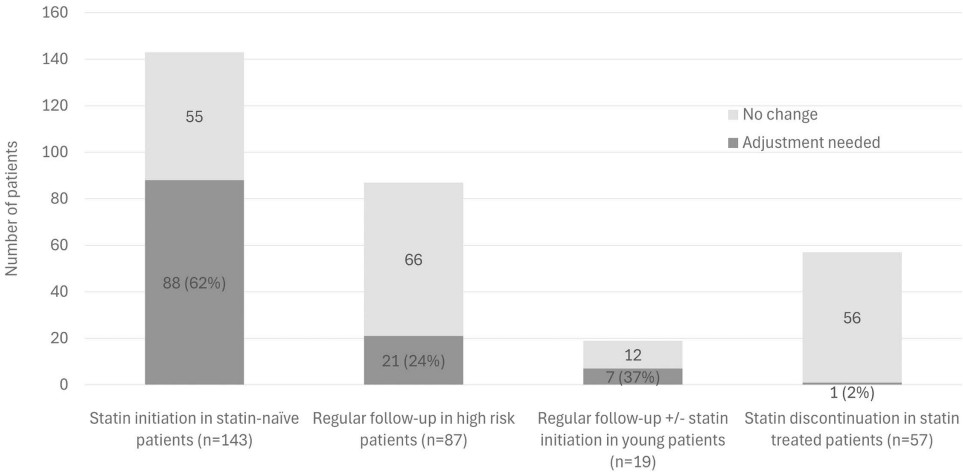

**Fig 3. Changes in primary CVD prevention in oncology patients.** This bar chart illustrates the potential impact of CAC assessment on CV management in oncology patients. Each column represents different subgroups: patients not receiving LLT; patients without prior cardiology consultation or CVD history; patients aged ≤ 45 years; patients receiving LLT. Dark grey sections represent patients in whom CAC assessment suggests a change in medical management. CAC, coronary artery calcium; CV, cardiovascular; CVD, cardiovascular disease; LLT, lipid-lowering therapy.

Classifying patients based on their CV risk remains challenging. Studies have demonstrated that CAC score is an effective tool for this purpose, as it provides a reliable assessment of atherosclerotic plaque burden and strong predictive value for medium-term CV events in both the general population and specific subgroups, such as young adults, women, or ethnic subgroups [21]. However, access to CAC scoring is limited due to the requirement for a dedicated, ECG-gated cardiac CT scan, resulting in few patients undergoing this test as part of primary prevention. Studies showed that assessing CAC is possible using low-dose non-gated CT acquisitions from PET/CT scans [22], with reliable results even on contrast-enhanced CT scans [23], which were used in 72% of the patients in our study. We observed excellent interobserver agreement for visual CAC scoring (κ = 0.90) despite the predominance of contrast-enhanced scans, in line with prior results [24,25]. Nevertheless, CAC score is rarely reported in routine practice following PET/CT imaging, as it requires specialized training for nuclear medicine physicians, and the clinical utility of this information in patient management remains uncertain.

Determining the CAC score in patients without regular cardiology follow-up or LLT could, however, be beneficial for many. Notably, there is a strong correlation between the qualitative visual assessment of CAC (absent, mild, moderate or severe), the quantitative CAC score and the risk of future CV events [26,27]. In this study, 43% of participants would potentially be reclassified – either upgraded or downgraded – in FRS category based on CAC scoring.

While all scientific societies agree that individuals with a CAC score ≥ 100, despite being classified as low-to-moderate risk, should be reclassified into a high-risk group for CV events and require regular follow-up, no international consensus exists regarding the therapeutic management based on CAC scores [28].

The ACC recommends withholding statin therapy for patients with a CAC score of 0 and no additional high-risk factors, consider statin therapy in CAC subgroups 1–99 for those aged ≥ 55 or those with higher-risk conditions and recommends statin therapy for all patients with a CAC score ≥ 100 [15].

In contrast, the European Society of Cardiology (ESC) suggests a different approach: a CAC score ≥ 100 may reclassify a patient to high risk, suggesting that statin therapy should be considered, depending on the patient's untreated low-density lipoprotein cholesterol levels. For CAC subgroups 1–99, the ESC does not recommend any specific therapeutic intervention [29].

**Table 3.  CAC according to clinical findings in breast cancer patients.**

| | |
|---|---|
| Number of patients | 58 |
| Age (years) | 62.8±12.9 |
| CV risk factors (n=52) | |
| Diabetes mellitus | 4 (6.9) |
| Hypertension | 24 (41.4) |
| Dyslipidemia | 15 (25.9) |
| Family history of CHD | 18 (31.0) |
| Smoking | |
| Never | 35 (60.3) |
| Previous | 17 (29.3) |
| Current | 6 (10.3) |
| Pack-years (n=23) | 13±12 |
| FRS (%) | 12.4±7.1 |
| History of CVD | 2 (3.4) |
| CAC | |
| CAC 0 | 13 (22.4) |
| Mild | 31 (53.4) |
| Moderate | 10 (17.2) |
| Severe | 4 (6.9) |
| Patients who had never consulted a cardiologist and had no history of CVD or invasive coronary angiography | 22 |
| CAC 0 | 5 (22.7) |
| Mild CAC | 13 (59.1) |
| Moderate CAC | 2 (9.1) |
| Severe CAC | 2 (9.1) |
| Patients not on LLT | 43 |
| CAC 0 | 13 (30.2) |
| Mild CAC, aged<55 and no higher-risk condition | 5 (11.6) |
| Mild CAC and aged≥55 or at least one higher-risk condition | 17 (39.5) |
| Moderate CAC | 6 (14.0) |
| Severe CAC | 2 (4.7) |

Data for continuous variables are presented as median (IQR) or mean±SD. Data for categorical variables are presented as number (percentage).

CAC, coronary artery calcium; CHD, coronary heart disease; CV, cardiovascular; CVD, cardiovascular disease; FRS, Framingham risk score; LLT, lipid-lowering therapy.

It is conceivable that in the future, additional pharmacological treatments may be recommended based on the CAC score. These could include intensification of LLT with statins and non-statins, as well as the introduction of aspirin or colchicine for the highest-risk patients [30].

Moreover, a recent meta-analysis showed a 55% overall increase in CVD mortality for patients with cancer, with risk varying across cancer types [31]. In this context the latest ESC guidelines on cardio-oncology recommend considering statin therapy for primary prevention in all patients with cancer at high or very-high cardiovascular toxicity risk. While the ESC does not explicitly endorse the use of CAC scoring to stratify CV risk in cancer patients, it acknowledges the role of CT imaging in detecting subclinical CVD, such as CAC, on routine imaging performed for cancer staging [32]. CAC

assessment may become a recommended practice before initiating anticancer therapy, supported by recent findings that show a strong correlation between CAC and the risk of cancer treatment-related CV toxicity, as evaluated using the specific Heart Failure Association–International Cardio-Oncology Society (HFA-ICOS) assessment tool [33]. According to ESC guidelines in onco-cardiology, patients should be referred to a cardiologist in various situations, including those with comorbidities, cardiac implantable electronic devices, baseline ECG abnormalities, preoperative assessment before oncologic surgery, initiation of cancer therapy with high or very-high CV toxicity risk, changes in cardiac symptoms, or evidence of CV toxicity based on clinical, biological, or radiological findings. Despite these recommendations, 44% of the patients in our study never had a consultation with a cardiologist. This can be explained by the difficult access to cardiology in under-medicalized areas. Adding CAC scoring to 18F-FDG PET could help identify cancer patients who should be prioritized for cardiology referral.

31% of the patients included in our study were women referred for breast cancer evaluation. Systematic assessment of CAC could be particularly valuable in these patients, as CV mortality has become the leading cause of death, surpassing cancer, in these patients [34]. This trend is attributed to advancements in cancer treatment, the associated increased CV risk linked to these therapies (e.g., anthracyclines, anti HER2 monoclonal antibodies, chest radiotherapy), overlapping initial risk factors (e.g., obesity, physical inactivity, alcohol consumption), and insufficient CV follow-up in women [35].

Our findings align with prior studies that have demonstrated the prognostic relevance of CAC scoring on non-gated, non-cardiac thoracic CT scans. A previous study already suggested the utility of CAC evaluation in breast cancer patients, demonstrating its superiority in predicting CV events compared to FRS. It also reported a high proportion of women with CAC (45%), an FRS exceeding 10%, and no statin therapy [36], in accordance with the findings of our study. Earlier studies also demonstrated that 18F-FDG uptake in major arteries was a strong predictor of a subsequent vascular event, whereas concomitant severe vascular calcifications seemed to impart a particularly high risk [37,38]. In a retrospective cohort of non-metastatic patients, CAC assessed on PET/CT was associated with mortality and cardiac events and statin prescriptions could have been modified in 60% of cases (84/140 patients) [39].

However, our study goes a step further by prospectively evaluating, in a real-world setting, the clinical impact of reporting CAC on PET/CT in oncology patients, not only in terms of risk stratification but also by correlating these findings with the patients' current CV therapy and cardiology follow-up status. This integrated approach allowed us to identify nearly 50% of patients for whom CAC findings would warrant therapeutic reclassification or follow-up adjustments based on current ACC guidelines. Unlike previous retrospective or imaging-focused studies, this prospective approach highlights the direct clinical applicability of CAC scoring on PET/CT. The method is simple, non-invasive, cost-free, does not involve additional radiation exposure, and could be readily implemented in routine practice to optimize primary CVD prevention in cancer patients — an increasingly recognized but under-addressed need in cardio-oncology.

Our study has several limitations. Its monocentric design may restrict the generalizability of the findings. The sample size, pragmatically set at 200 patients, was insufficient for multivariable analyses, raising the possibility of residual confounding.

The reliance on self-administered questionnaires likely introduced reporting and recall bias, particularly for socially sensitive behaviors such as smoking, as well as for family history of CHD, which may have been over-reported. These biases may partly explain their lack of association with CAC.

Additionally, participation depended on patients' willingness and ability to complete the questionnaire, potentially leading to selection bias with underrepresentation of certain subgroups, such as patients with cognitive impairment, language barriers, or severe clinical conditions.

Methodologically, the use of contrast-enhanced CT for CAC scoring, although based on validated approaches [23,24], and increasingly widespread, remains imperfect and may have underestimated low levels of calcification, thereby underestimating the prevalence of CAC and the number of patients eligible for preventive interventions such as LLT.

Finally, the cross-sectional design precludes evaluation of longitudinal outcomes. As such, the prognostic implications of CAC scoring for incident cardiovascular events cannot be inferred.

## Conclusions

In our study, CAC score on 18F-FDG PET imaging identified 49% of the patients who would benefit from systematic scoring in their cardiovascular management, including being reclassified as high risk of CVD, warranting a regular follow-up, and/or LLT adjustments, in accordance with ACC recommendations. CAC can be visually assessed and reported on 18F-FDG PET/CT, even with contrast enhancement.

These findings suggest that a systematic evaluation of CAC on PET/CT imaging could be valuable, particularly to improve cardiovascular risk stratification and CVD prevention in oncological patients. Whether this optimization of primary CVD prevention in cancer patients translates into clinical benefit remains to be evaluated.

## Supporting information

**S1 Table. Comparison of median CAC scores between patient subgroups using the Wilcoxon rank-sum test.**
CAC, coronary artery calcium; CHD, coronary heart disease; CVD, cardiovascular disease.
(DOCX)

**S1 Fig. Moderate or severe CAC according to PET/CT indication.** This bar chart illustrates the proportion of patients with moderate to severe calcifications (dark grey), indicating a high cardiovascular risk according to the CAC score, within each patient subgroup defined by cancer type. The proportion is significantly higher in the "lung cancer" subgroup ($p < 0.05$). CAC, coronary artery calcium.
(TIF)

## Acknowledgments

We express our deepest gratitude to the cardiologists, the technical and radiopharmaceutical staff from CHU Orléans. This study is part of the French network of University Hospitals HUGO ('Hôpitaux Universitaires du Grand Ouest').

## Author contributions

**Conceptualization:** Mathilde Wanneveich, Denis Angoulvant, Matthieu Bailly.

**Data curation:** Rejane MAZET.

**Formal analysis:** Mathilde Wanneveich.

**Investigation:** Rejane MAZET, Gilles METRARD.

**Methodology:** Mathilde Wanneveich.

**Project administration:** Matthieu Bailly.

**Resources:** Nouritza Torossian.

**Software:** Gilles METRARD.

**Supervision:** Denis Angoulvant.

**Validation:** Matthieu Bailly.

**Writing – original draft:** Rejane MAZET.

**Writing – review & editing:** Nouritza Torossian, Mathilde Wanneveich, Gilles METRARD, Angela Malmare, Denis Angoulvant, Matthieu Bailly.

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
