## [Decision Letter · Decision Letter 0]

19 Aug 2025

Dear Dr. Bailly,

Thank you for submitting your manuscript to PLOS ONE. After careful consideration, we feel that it has merit but does not fully meet PLOS ONE’s publication criteria as it currently stands. Therefore, we invite you to submit a revised version of the manuscript that addresses the points raised during the review process. Please ensure that your decision is justified on PLOS ONE’s publication criteria  and not, for example, on novelty or perceived impact.

We look forward to receiving your revised manuscript.

Kind regards,

Gaetano Santulli, MD, PhD, FAHA

Academic Editor

PLOS ONE

2. Please note that your Data Availability Statement is currently missing [the repository name and/or the DOI/accession number of each dataset OR a direct link to access each database]. If your manuscript is accepted for publication, you will be asked to provide these details on a very short timeline. We therefore suggest that you provide this information now, though we will not hold up the peer review process if you are unable.

Reviewers' comments:

Reviewer's Responses to Questions

**Comments to the Author**

1. Is the manuscript technically sound, and do the data support the conclusions?

Reviewer #1: Yes

Reviewer #2: Yes

2. Has the statistical analysis been performed appropriately and rigorously?

Reviewer #1: Yes

Reviewer #2: Yes

3. Have the authors made all data underlying the findings in their manuscript fully available?

Reviewer #1: Yes

Reviewer #2: No

4. Is the manuscript presented in an intelligible fashion and written in standard English?

Reviewer #1: Yes

Reviewer #2: Yes

Reviewer #1: The manuscript is very interesting and provides an innovative useful tool in addition to PET scan reporting.

I asked only to modify/add few things.

1- I notice that some abbreviations are missed in the corrispondent paragraph in page 3, for example LLT and CHD. Then in page 2 you use CAC as 'coronary artery calcification' instead as 'coronary artery calcium' (page 3), could you use the same abbreviation?

2- In page 9 line 179 you report that one patient can't have the quantitative CAC scoring: why? if you think it could be interesting, you can add this info.

3- In page 10 line 190-196 you correlate the CAC score with gender without specification of the age: there is some differences in younger/older male or female? If you have a significant difference, could you add in the description. Moreover, could you divide your patients in ethnic subgroups or are all of the same subgroup (please add the info)?

4- The same in page 13-14-15: It seems, as you describe in your report, that only in lung and breast cancer is important to use the CAC score; if it's not what you mean (and I think so), you could describe the other tumours in which you find a significance, it could be really interesting for physician who work in oncological institutes.

5- Page 16 line 277: why do you think there is an incosistent self-reporting about smoking in some patients? I think it's something you can't calculate, and obviously you have to believe in what they report you; are there some factors that could let you think so? Or do you have any scientific works that tell you this (please add in the biblio)?

Good work!

Reviewer #2: Dear Editore and Authors,

I have read with interest this manuscript.

Find my comments listed below.

Major Comments

1. Several prior studies have investigated incidental CAC scoring on non-gated CT scans, including in cancer patients. Please clarify explicitly what makes this study novel.

2. The study was conducted over a short time window (June–July 2024) and at a single center. Please specify whether all consecutive patients were enrolled, and discuss potential selection bias.

3. Provide justification for the sample size. Was a power calculation performed? If not, please acknowledge this limitation.

4. Since 72% of CT scans were contrast-enhanced, the methodology for adjusting HU thresholds is crucial. Please provide more details on validation of this method, and discuss its limitations in greater depth (e.g., risk of underestimating CAC at low levels of calcification).

5. The manuscript states that visual and automated scoring showed no significant difference. Please provide interobserver agreement data to strengthen reliability.

6. The reclassification into different CVD risk categories should be more explicitly described. Did you strictly apply the 2019 ACC/AHA guidelines? Were ESC guidelines also considered? Please detail the exact thresholds and rules used.

7. The analyses are primarily descriptive. Consider performing multivariable regression to identify independent predictors of moderate/severe CAC in this population. This could add depth to your findings.

8. Statistical analyses. Please clarify whether adjustments for multiple comparisons were applied.

9. Emphasize that the clinical consequences remain hypothetical and need validation in outcome-driven studies.

10. Limitations. Explicitly mention the lack of longitudinal cardiovascular outcomes as a limitation of the study.

11. Limitations. Acknowledge potential recall bias from self-reported risk factors.

12. Only oral informed consent was obtained. Please clarify why written consent was not required, and whether this was explicitly approved by the ethics committee. This is especially important given the prospective trial registration.

13. References. The reference list is adequate but could be updated with more recent cardio-oncology consensus documents.

Minor Comments

There are some grammatical issues in the manuscript. An English editing is recommended.

Table 1 is dense; consider restructuring into a more reader-friendly format.

Figures 2 and 3: please enlarge labels and provide more informative legends.

Ensure that all abbreviations are defined at first mention in each table/figure.

References. Please check formatting consistency (page ranges, punctuation).

**Do you want your identity to be public for this peer review?** For information about this choice, including consent withdrawal, please see our Privacy Policy

Reviewer #1: No

Reviewer #2: No

---

## [Author Response · Author response to Decision Letter 1]

2 Sep 2025

Dear Editor and Reviewers.

Thank your for reviewing our manuscript. Please find below our answers to your comments.

Reviewer #1: The manuscript is very interesting and provides an innovative useful tool in addition to PET scan reporting.

I asked only to modify/add few things.

1- I notice that some abbreviations are missed in the corrispondent paragraph in page 3, for example LLT and CHD. Then in page 2 you use CAC as 'coronary artery calcification' instead as 'coronary artery calcium' (page 3), could you use the same abbreviation?

Thank you for pointing this out. We have carefully revised the manuscript to ensure consistency in the use of abbreviations.

2- In page 9 line 179 you report that one patient can't have the quantitative CAC scoring: why? if you think it could be interesting, you can add this info.

Quantitative CAC scoring could not be performed for one patient who was unable to maintain a strict supine position during image acquisition, which prevented the use of the Syngo. CT CaScoring software. We agree that this information may be of interest and have added this clarification to the manuscript.

3- In page 10 line 190-196 you correlate the CAC score with gender without specification of the age: there is some differences in younger/older male or female? If you have a significant difference, could you add in the description. Moreover, could you divide your patients in ethnic subgroups or are all of the same subgroup (please add the info)?

Regarding the potential age difference between male and female patients, we performed a Student's t-test and found no statistically significant difference in age between the two groups:

• Male patients: 66.2 ± 13.6 years

• Female patients: 64.2 ± 13.1 years

• p-value: 0.33

Therefore, the correlation between CAC score and gender was not confounded by age in our cohort.

Regarding ethnicity, most patients probably belonged to the same ethnic group (Caucasian). However, ethnic categorization is generally not realized in our country due to legal and constitutional restrictions. According to Article 6 of the French Data Protection Act (Law No. 78-17 of January 6, 1978, “Informatique et Libertés”), it is prohibited to process personal data revealing racial or ethnic origin, political opinions, religious or philosophical beliefs, except under specific scientific study exceptions. Although such exceptions exist, collecting or reporting ethnic data is rarely done in practice.

We have therefore not performed detailed ethnic subgroup analyses in this study.

4- The same in page 13-14-15: It seems, as you describe in your report, that only in lung and breast cancer is important to use the CAC score; if it's not what you mean (and I think so), you could describe the other tumours in which you find a significance, it could be really interesting for physician who work in oncological institutes.

Thank you for this relevant comment.

As mentioned in the manuscript, we did explore potential differences according to tumor type. We found that the proportion of patients classified as high risk of cardiovascular disease was particularly elevated in the lung cancer group. Additionally, CAC scoring had a particularly notable impact on primary cardiovascular prevention strategies in the breast cancer subgroup — a finding of clinical importance, as these patients tend to be undertreated, as discussed in the manuscript.

While we aimed to group cancer types into a limited number of meaningful categories, the sample sizes for other tumor groups were too small to allow robust subgroup analyses or draw statistically significant conclusions.

Nevertheless, this does not diminish the relevance of our findings. Importantly, regardless of the primary cancer type, we observed that CAC scoring led to suggest a change in cardiovascular management in nearly one out of two patients, reinforcing the broad utility of this approach across oncologic indications.

5- Page 16 line 277: why do you think there is an incosistent self-reporting about smoking in some patients? I think it's something you can't calculate, and obviously you have to believe in what they report you; are there some factors that could let you think so? Or do you have any scientific works that tell you this (please add in the biblio)?

Several studies have demonstrated a clear association between CAC prevalence and history of smoking (PMID: 24468147; 23500326; 33879501)

However, in our study, we did not observe a statistically significant difference in CAC scores between smokers (former or current) and non-smokers. This may be due to potential information bias related to self-reported smoking status.

We agree that this information may be of interest and have added this clarification to the manuscript.

Reviewer #2: Dear Editore and Authors,

I have read with interest this manuscript.

Find my comments listed below.

Major Comments

1. Several prior studies have investigated incidental CAC scoring on non-gated CT scans, including in cancer patients. Please clarify explicitly what makes this study novel.

While prior studies only focused on the description of CAC scoring on non-gated CT, most often in retrospective cohorts, our study is prospective, real-world, that not only assesses CAC scoring on oncologic PET/CT but also correlates it with each patient's actual cardiovascular management—including lipid-lowering therapy and cardiology follow-up.

We clarified this information in the revised manuscript.

2. The study was conducted over a short time window (June–July 2024) and at a single center. Please specify whether all consecutive patients were enrolled, and discuss potential selection bias.

All patients referred to our center for an 18F-FDG PET/CT scan from June 2024 onward were given the opportunity to participate in the study, based on their willingness to do so. Enrollment was proposed systematically to each eligible patient during the study period (June–July 2024), regardless of clinical indication, ensuring consecutive recruitment in routine clinical practice.

As noted, a potential source of selection bias is the exclusion of patients unable to complete a self-administered questionnaire (e.g., due to cognitive impairment, language barriers, or severe clinical condition), which may have led to underrepresentation of certain subgroups. This limitation has been acknowledged and is now discussed in the revised manuscript.

3. Provide justification for the sample size. Was a power calculation performed? If not, please acknowledge this limitation.

As this is a descriptive, exploratory study, a formal power calculation was not performed, which represents a limitation of the work. Determining the required sample size in such contexts is inherently challenging.

A recent study assessed the coronary artery calcium (CAC) score on chest CT scans in the context of lung tumor management. A total of 193 patients were included; 22% had known cardiovascular disease, and 74% showed coronary calcifications. Several cardiovascular and cancer risk factors are shared (such as smoking, obesity, hyperglycemia, physical inactivity), which may explain these findings (PMID: 36531700)

We thought that our study could yield similar results but aimed to investigate a broader range of imaging indications. The total number of patients expected to be included in this study was 200. This number was consistent with the recruitment capacity of the department (>6,500 patients undergoing PET/CT scans over 12 months).

4. Since 72% of CT scans were contrast-enhanced, the methodology for adjusting HU thresholds is crucial. Please provide more details on validation of this method, and discuss its limitations in greater depth (e.g., risk of underestimating CAC at low levels of calcification).

Several studies have validated the methodology of quantifying coronary artery calcification (CAC) from contrast-enhanced CT scans (PMID: 23943125; 35475394)

These studies demonstrate that contrast-enhanced CT provides high diagnostic performance for the presence of CAC compared to noncontrast scans. For example, one study reported sensitivity of 83% versus 89% (p = 0.20), specificity of 100% for both (p = 0.99), positive predictive value (PPV) of 100% for both (p = 0.99), and negative predictive value (NPV) of 76% versus 83% (p = 0.34) for contrast-enhanced versus noncontrast CT, respectively.

However, it is important to acknowledge limitations. The use of contrast enhancement may lead to underestimation of CAC burden, particularly at low levels of calcification

We have expanded the discussion in the manuscript to address these limitations and their potential impact on our results. However, the use of contrast-enhanced CT on PET/CT is increasing, thus we thought it important to include it as part of this real-life work.

5. The manuscript states that visual and automated scoring showed no significant difference. Please provide interobserver agreement data to strengthen reliability.

Thank you for this valuable comment. We observed excellent interobserver agreement for visual CAC scoring (κ = 0.90), which is consistent with values reported in previous studies (PMID: 21109114; 35475394). This information has now been added to both the Results and Discussion sections to reinforce the reliability of our findings.

6. The reclassification into different CVD risk categories should be more explicitly described. Did you strictly apply the 2019 ACC/AHA guidelines? Were ESC guidelines also considered? Please detail the exact thresholds and rules used.

We reclassified patients based on their CAC scores according to the 2019 ACC/AHA guidelines as follows:

• Patients with a CAC score of zero were classified as low risk.

• Patients with CAC scores >100 Agatston units (moderate to severe calcifications) were classified as high risk.

• Patients with CAC scores between 1 and 99 (mild calcifications) were classified as intermediate risk.

Our classification strictly followed the ACC/AHA 2019 recommendations, excerpts of which include:

• “Coronary artery calcium measurement can reclassify risk upward (particularly if coronary artery calcium score is ≥100 Agatston units (AU) or ≥75th age/sex/race percentile) or downward (if coronary artery calcium is zero) in a significant proportion of individuals.”

• “The absence of coronary artery calcium could reclassify a patient downward into a lower risk group in which preventive interventions (e.g., statins) could be postponed.”

• “For those with coronary artery calcium scores of 1 to 99 AU, 10-year ASCVD event rates are 3.8%, 6.5%, and 8.3% for adults 45 to 54, 55 to 64, and 65 to 74 years of age, respectively, indicating that risk reclassification is modest for individuals with coronary artery calcium scores of 1 to 99. Therefore, for patients with coronary artery calcium scores of 1 to 99, it is reasonable to repeat the risk discussion.”

ESC guidelines offer a less detailed framework than ACC/AHA guidelines. They only recommend classifying patients with a CAC score >100 AU as high risk, without providing specific guidance on the management of patients with lower CAC scores or the management of lipid-lowering therapy according to CAC score.

We have now added these details and references explicitly in the revised manuscript.

7. The analyses are primarily descriptive. Consider performing multivariable regression to identify independent predictors of moderate/severe CAC in this population. This could add depth to your findings.

Our sample size was relatively small (n=190), although comparable to similar studies. This allowed us to identify significant differences when comparing various subgroups, after adjustment for multiple comparisons. However, it limited the feasibility of regression analyses with multiple covariates.

We attempted to perform a linear regression analysis using the continuous CAC score as the dependent variable, but the model assumptions were not met, even after several attempts at data transformation. Alternatively, an ordinal logistic regression could theoretically be applied to distinguish CAC categories (no, mild, moderate, severe). However, this approach would require a larger sample size given the number of predictors to be included and the unbalanced distribution of CAC categories in our cohort (24.7% no CAC, 36.8% mild, 21.6% moderate, and 16.8% severe).

8. Statistical analyses. Please clarify whether adjustments for multiple comparisons were applied.

We performed adjustments for multiple comparisons using Benjamini-Hochberg method.

These changes have been incorporated into the revised manuscript.

9. Emphasize that the clinical consequences remain hypothetical and need validation in outcome-driven studies.

10. Limitations. Explicitly mention the lack of longitudinal cardiovascular outcomes as a limitation of the study.

11. Limitations. Acknowledge potential recall bias from self-reported risk factors.

Thank you for these valuable comments. We have revised the discussion to acknowledge these limitations.

12. Only oral informed consent was obtained. Please clarify why written consent was not required, and whether this was explicitly approved by the ethics committee. This is especially important given the prospective trial registration.

In France, a RIPH 3 study is a low-risk, non-interventional study in which all medical procedures and treatments are performed as part of routine care, without any additional or unusual interventions. This category includes studies where the only intervention is a self-administered questionnaire completed by the patient.

Patients can be included by a licensed physician or a qualified medical trainee. Regulatory requirements include:

• Providing participants with an information sheet

• Approval by an Ethics Committee (CPP)

• Registration on clinicaltrials.gov

• Registration with the French Data Protection Authority (CNIL)

This category allows research with minimal risk, ensuring patient safety and data protection without extra procedures.

The ethics committee approved our study protocol as described below:

“The study information sheet was sent to patients along with their appointment letter for the PET/CT scan. Then, on the day of the examination (on average between 24 hours and one week later), participation in the study was proposed to patients in our nuclear medicine department by a qualified investigator. The investigator provided the patient with clear and comprehensive information regarding the study procedures, objectives, and participants’ rights in the context of clinical research. The investigator was available to answer any questions the patient might have.

The self-administered questionnaire was completed by the patient and returned to one of the radiology technologists or to a qualified investigator before leaving the department.

Patients could decline participation in the study at any time before leaving the department, or later by making an explicit request.”

13. References. The reference list is adequate but could be updated with more recent cardio-oncology consensus documents.

Thank you for this comment. We have updated the reference list to include more recent cardio-oncology consensus documents.

Best Regards,

---

## [Editor Report · Decision Letter 1]

8 Oct 2025

Calcium scoring during 18F-FDG PET/CT in cancer indications: improving cardiovascular risk stratification and prevention

PONE-D-25-37964R1

Dear Dr. Bailly,

We’re pleased to inform you that your manuscript has been judged scientifically suitable for publication and will be formally accepted for publication once it meets all outstanding technical requirements.

Kind regards,

G. Santulli

Academic Editor

PLOS ONE

---

## [Editor Report · Acceptance letter]

PONE-D-25-37964R1

PLOS ONE

Dear Dr. Bailly,

I'm pleased to inform you that your manuscript has been deemed suitable for publication in PLOS ONE. Congratulations! Your manuscript is now being handed over to our production team.

Kind regards,

on behalf of

Professor Gaetano Santulli

Academic Editor

PLOS ONE